# Macroscopic and microscopic study on floral biology and pollination of *Cinnamomum verum* Blume (Sri Lankan)

**Bhagya M. Hathurusinghe**[1], **D. K. N. G. Pushpakumara**[2], **Pradeepa C. G. Bandaranayake**[1]*

1 Agricultural Biotechnology Centre, Faculty of Agriculture, University of Peradeniya, Peradeniya, Sri Lanka,
2 Department of Crop Science, Faculty of Agriculture, University of Peradeniya, Peradeniya, Sri Lanka

* pradeepag@agri.pdn.ac.lk

**Data Availability Statement:** All relevant data are within the paper and its Supporting information files.

## Abstract

*Cinnamomum verum* Blume (syn *Cinnamomum zeylanicum*) commonly known as Ceylon cinnamon, has gained worldwide attention due to its health benefits and its unique quality. Therefore, maintaining the yield quality and quantity is essential, especially for high-end value-added products. Knowledge on floral behaviour and reproductive biology is essential for breeding superior varieties and is critical for commercial cultivation efforts. However, limited literature is available on the floral biology of *C. verum*. Here in this study, we assessed the seasonal flowering, floral development and pollination of two cultivars of *C. verum*. Both macroscopic and microscopic data were collected on floral biology, pollination, and male and female floral organs before and after pollination. *Cinnamomum verum* is morpho-anatomically, structurally, and physiologically adapted for cross-pollination, possible between the two cultivars; type A (*Sri Gemunu*) and type B (*Sri Wijaya*) flowers; naturally evolved with Protogynous Dichogamy. However, due to changes in environmental conditions, female and male stages in the same tree overlap for about 45–60 min suggesting possible close-pollination within the same plant. During this event some of the pollens were observed hydrated even during self-pollination. In mean time, 4–8% of the flowers formed fruits after natural close and hand pollination which is between male and female phases of the same tree. Although *C. verum* is adapted for cross-pollination, natural close-pollination is also possible. The data suggest the complex nature of the sexual reproduction of *C. verum*. Well-managed breeding attempts with controlled factors like temperature and humidity will help to develop superior *C. verum* varieties.

## Introduction

The Genus *Cinnamomum* Scharffers belongs to the family Lauraceae, which comprises over 250 species [1] distributed from the Asiatic mainland to Formosa, the Pacific islands, Australia, and Tropical America [2]. *Cinnamomum* is a pantropical genus consisting of evergreen trees and shrubs [3]. While some economically important species such as *C. camphora* J. Prei and *C.*

**Funding:** The Ministry of Primary Industries and Social Empowerment through the National Science Foundation of Sri Lanka under the special Cinnamon project – Grant No: NSF SP/CIN/2016/01 and the Early Career Fellowship of the Organization for Women in Science for Developing countries (OWSD). The funders had no role in study design, data collection and analysis, decision to publish, or preparation of the manuscript.

**Competing interests:** The authors have declared that no competing interests exist.

*aromaticum* Nees are trees, others such as *C. verum*, *C cassia*, *Cinnamomum burmannii*, (Indonesian cinnamon), and *Cinnamomum loureiroi* (Vietnamese cinnamon) [4], and *Cinnamomum tamala* T.Nees & C.H.Eberm., (India and Nepal) [5] are shrubs.

All the *Cinnamomum* species recorded so far are cross-pollinated with Synchronised protogynous dichogmay as the prominent mechanism [6, 7]. Diachogamy is defined as the temporal separation of male and female functions. In protandrous species, another dehiscence occurs before the stigma becomes receptive, while it is reversed in protogynous species [8]. Here, the same flower opens twice, either in the morning or in the afternoon of the first day, and in the afternoon or the morning of the next day, respectively [9]. The same phenomenon has been recorded in *C. verum* in Sri Lanka [10].

The cinnamon flower has both functional male and female organs [11], and there are two types of varieties. In the type-A varieties, female flowers open for the first time in early or mid-morning, remain open and pistil receptive until about noon, then close and remain closed until noon of the following day, when they reopen and begin shedding pollen with the pistil no longer receptive. The flowers close permanently in the late evening, after a cycle of about 36 hours. The flowers of type B varieties function analogously but with transposed timing. The opening cycle of type B flowers spans about 24 hours, and the difference in cycle time reflects the relative length of the closed period between openings [11]. Therefore, cross-pollination is possible with type A and type B varieties.

Avocado, one of the other economically important species in the family Lauraceae, also evolved with Protogynous Dichogamy breeding system [12]. The Protogynous dichogamy of avocado presents two morphs for successful cross-pollination and flowering is complementarily synchronized. In contrast to the biological adaptation, occurrence of self-pollination in orchards composed of a single genotype probably due to some overlapping between male and fame stages in the same flower, among flowers of the same tree also has been recorded [13]. The flowering cycle of protogynous dichogamy plants are extremely sensitive to environmental conditions [14, 15], where flowering behaviour fluctuates with temperatures and humidity [12, 16].

Davenport in 1986 [17] defined the morphological development of the avocado flower, relating to the histological details. However, such information on *Cinnamomum* is limited.

Considering cinnamon, since vegetatively propagated planting materials create crop management issues in largescale cultivations [18, 19], seedlings are mostly preferred by growers. Therefore, studying the possible selfing is essential for producing quality planting materials from known superior varieties and for future breeding attempts. However, lack of ample amount of knowledge on the floral biology of cinnamon is one of the major drawbacks of missing required characters for hybridization. With the hypothesis that the same self-pollination within the same species is possible, the following study was carried out. Lack of ample amount of knowledge on floral biology of cinnamon is one of the major drawbacks of missing required characters for hybridization. Against this backdrop, the current study focused on macroscopical and microscopical changes in type A and type B flowers of cinnamon and possible self-pollination of both types.

A closer look at floral biology, pollination, and fertilization would facilitate such efforts. Therefore, the current study focused on the detailed floral biology and morphology of selected type A and type B of *C. verum* varieties before and after pollination.

## Materials and methods

### Study site and sample collection

The experiments were conducted for two consecutive years (2019–2020) in a vegetatively propagated orchard located at the Delpitiya sub-research station of the Department of Export

Agriculture, Atabage, Sri Lanka (707.970'N, 80035.342'E; 634m MSL) and greenhouse at the Agricultural Biotechnology Centre, Peradeniya, Sri Lanka. The experimental field consisted of two cultivars.

*Sri Gemunu* (Type A) and *Sri Wijaya* (Type B), in alternative rows. Randomly selected six plants in orchard at Atabage, Sri Lanka with flowers from each cultivar marked and selected as replicates. The flowering phenology and floral behaviour were monitored from 7.30 am to 4.00 pm. The meteorological data viz., minimum, and maximum temperature and relative humidity of Delpitiya were recorded. The temperature was recorded hourly while the humidity was recorded two times a day.

## Flower and inflorescence morphology

Since flowers are the resources for the pollinators, the morphology or the outlook place a great role during pollination. Therefore, key morphological attributes of flowers and inflorescences were measured to characterize the visual display and the resource accessibility of the reproductive units. The length of flowering inflorescences and the number of inflorescences was determined in the field on selected individuals with a ruler. The number of flowers in inflorescences and the number of inflorescences in inflorescence clusters were counted in the field. Three selected inflorescences clusters were tagged. We recorded the number of opened and closed flowers in an inflorescence, flowering period, and flower lifespan, and measured the length of the stigma, petal, stamen, and corolla taking the mean values of 10 replicates.

Floral morphology was studied with fresh flowers using a Stereomicroscope (Kruss Opironic and Olympus SZX10). The images were captured from the software AmScope V/ 3.7.2776 and Olympus Cell Sens Standard V 1.16). Dimensions of single floral organs and their position within the flower were determined with size differenced micrographs using the ImageJ software [20].

## Dynamics of flowering

Randomly selected six trees from both cultivars, *Sri Gemunu* and *Sri Wijaya*, were taken to study the flowering dynamics. We collected the flower phenology data by counting the number of flowers open in each selected plant during both phases. The length of the flowering period was defined as the number of days between the opening of the first flower and the closing of the last flowers. It was measured for two years in 2019 and 2020, during the mass flowering season, January-February. Flowers were considered open if pollinators could use their petals at least 45 degrees open and closed if both the stigmas and anthers wilted. The flowering progress was monitored at 30-minute intervals every day. We used the phenology data collected to determine the mass or peak flowering season for each cultivar, flowering period, and overlapping period. The flowering of labelled inflorescence was observed for 18 days during flowering seasons before flowering, with 25% of flowers open, peak flowering >50% open and termination of flowering. To determine the flowering periods, the number of pollination units with opened flowers in each plant was labelled and counted.

**Flowering synchrony index.** Flowering synchrony was measured using an index that considers the proportion of open female flowers on the focal plant that overlapped temporally with the open male flowers in the next phase. The synchrony index (SI) was calculated with the equation given [21];

$$SI_p = 1 - 0.5 \sum_{d=I}^{D} \left| \frac{F_{pd}}{F_{ps}} - \frac{F_{nd}}{F_{ns}} \right|$$

The index outputs values from zero to one, where one is the most synchronous and indicial could be and zero is not synchronous at all. Synchrony index was counted for the flowers that were open during the overlap period. where $F_{pd}$ and $F_{nd}$ are the numbers of flowers open on focal plant $p$ and all its neighbours $n$, respectively, on day $d$; $D$ is the number of days those flowers were counted for an individual; and $F_{ps}$ and $F_{ns}$ are the total numbers of flowers produced during the season by focal plant $p$ and all of its neighbours $n$, respectively.

### Pollen-stigma interaction

We conducted controlled experimental pollinations to understand the possibility of natural selfing in *C. verum*. Tis experiment was carried out on premises at Agricultural Biotechnology Centre. Three plants of *Sri Gemunu* were isolated in separate net houses made from shade nets/mesh (Hole size 1 mm x 2 mm). They were built to avoid the possibility of pollination from the neighbour plant and to avoid any external influence on pollination and kept at a considerable distance (Fig 1).

Out of the three plants, two were allowed to close-pollinate via ants, which was identified as an efficient pollinator in the field, while another one was close-pollinated by hand. Further, five (5) pistils from each pollinated flower at different time intervals (after 1 hour, 4 hours, and 20 hours of opening) were collected to observe the growth of the pollen tube. Pistils were immediately subjected to fixing solution (acetic acid: absolute alcohol, 1:3) overnight and then transferred to70% ethanol and preserved.

To observe the pollen and stigma interaction, the flowers were harvested from three plants at three stages, i. female flowers just opened before being stigma receptive, ii. female flowers after stigma being receptive iii. female flowers after being pollinated, and iv. pollinated male flowers after anthesis. Three individual flowers from each were selected and visibly observed pollination by the pollinators after 3–5 hours of observation.

The structural and morphological variation in the stigma which were subjected to pollinated during overlapping period was observed before and after pollination in both '*Sri Wijaya*' and '*Sri Gemunu*', using a Scanning Electron Microscope. Further stigma-pollen interaction and self (in)compatibility were observed using a fluorescence microscope.

### Scanning Electron Microscopy, sample preparation and imaging

The pollen and stigma of the collected specimens were studied using Scanning Electron Microscopy (SEM). For SEM analysis, the flowers at the anthesis stage and pollinated stigma were used. The fresh anthers and stigma were separated from the flower and fixed in FAA solution, 5% formaldehyde (v/v), 5% (v/v) acetic acid, and 45% (v/v) ethanol for long-term preservation.

The preserved fresh pollen samples were centrifuged at 2,500 rpm for 3 minutes. The pellet was then washed twice with distilled water for 25–30 minutes. After the washing, pollens were dehydrated in an ethanol series (100% DW→3:1→1:1→3:1→100% ethanol) for about 15 minutes at each stage. The pollens were separated by centrifugation at 2500 rpm for 3 minutes at each step, after making sure that the ethanol and water are removed. For specimen drying, the samples were then transferred into 100% Hexamethyldisilazane (HMDS) through a graded series (100% ethanol→3:1→1:1→3:1→100% HMDS) and soaked for about 20 minutes at each stage. Then pollen was air-dried until all HMDS get evaporated. The processing of the preserved stigmas followed the same chemical treatments without centrifugation steps.

The samples were mounted onto the sample stub using carbon tapes and gold sputter-coated for 15 seconds. The coated samples were examined by Hitachi SU6600 Analytical Variable Pressure FE-SEM (Scanning Electron Microscope) at Sri Lanka Institute 0f Nanotechnology (SLINTEC), Homagama, Sri Lanka.

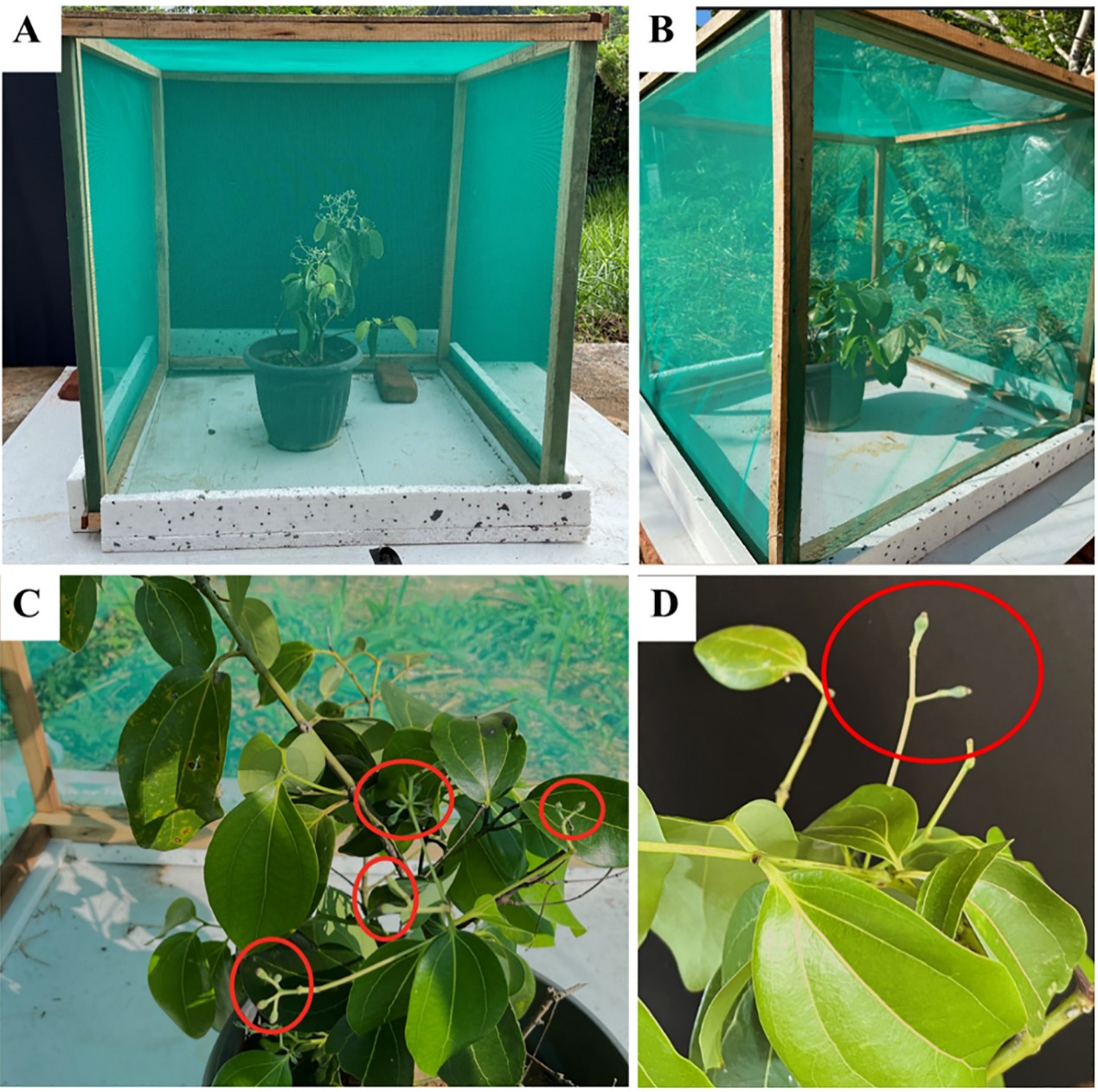

**Fig 1. Field experimental blocks for observing the efficiency in close pollination located at Agricultural Biotechnology Centre.** (A) Net house built for isolation of cinnamon plants for insect pollination and hand pollination. Insect pollination is from ants (B) Net house built for isolation of cinnamon plants for hand pollination. hand pollination was done between two female and male flowers from the same tree, during the overlapping period (C) fruit production yielding 8% of total flowers produced, pollination method: hand pollination (D) fruit production yielding 4% of total flowers produced, pollination methods: insect pollination.

### Fluorescence microscopy

We followed the protocol described by Dafni in 1992 [22]. For the observation of pollen tubes growing inside the style, the fixed excised stigma and style in FAA were treated with 70% and then washed with distilled water. Fixed stigma samples were kept at +42˚C in the refrigerator until staining. Pistils were softened by keeping them in 8 M NaOH at 60˚C for 4h. They were

then rinsed in running water for 3 hours to remove NaOH. Then stained in 0.005% aniline blue in potassium acetate for 6 h and mounted in a 1:1 v/v mixture of aniline blue and glycerin. The slides were observed under a fluorescence microscope with a UV filter and the germination of pollen on the stigma and path of pollen tubes through the stylar tissue to the ovule.

### Statistical analysis

The total flowers opened and overlapped during both peak season and off-season were analysed for normality using Sharpiro-Wilk normality test before selecting parametric or non-parametric variations of each statistical test. Variances of data distribution were compared using F-tests. Pearson correlation analysis was performed to investigate any significant correlation among the floral morphological traits. One-way analyses of variance were used to test differences among the populations, and their significance was tested through Fisher's least significance difference (LSD). Further, Synchrony index was calculated to identify the synchrony for phenological events of *Sri Gemunu* and *Sri Wijaya* within Peak and off-peak season. As it was significant and overlapping was prominent during peak season, only data collected during peak season were considered for the analysis. To check the Correlation between climatic factors and overlapping probability, Pearson analysis was carried out to identify the significance of climatic factors; temperature and humidity for the overlapping oof *Sri Gemunu* and *Sri Wijaya* stages. Fit-Regression Model was used to identify the how the weather parameters impact on the overlapping of male and female stages in two cultivars All analyses were done in SAS studio version 3.8 and Analyse-it Standard Version 2.30 and Minitab version 17 (State College, PA: Minitab, Inc).

## Results

### Floral biology

*Cinnamomum verum* flowers are seen as clear to yellowish-green. Tepals on the outside are greenish-white oblong-lanceolate, tepals on the interior are pale white (Figs 2A and 3), and newly opened flower nectaries are yellow (Fig 2A and 2B). There were no distinct floral morphology differences between *Sri Gemunu* and *Sri Wijaya*.

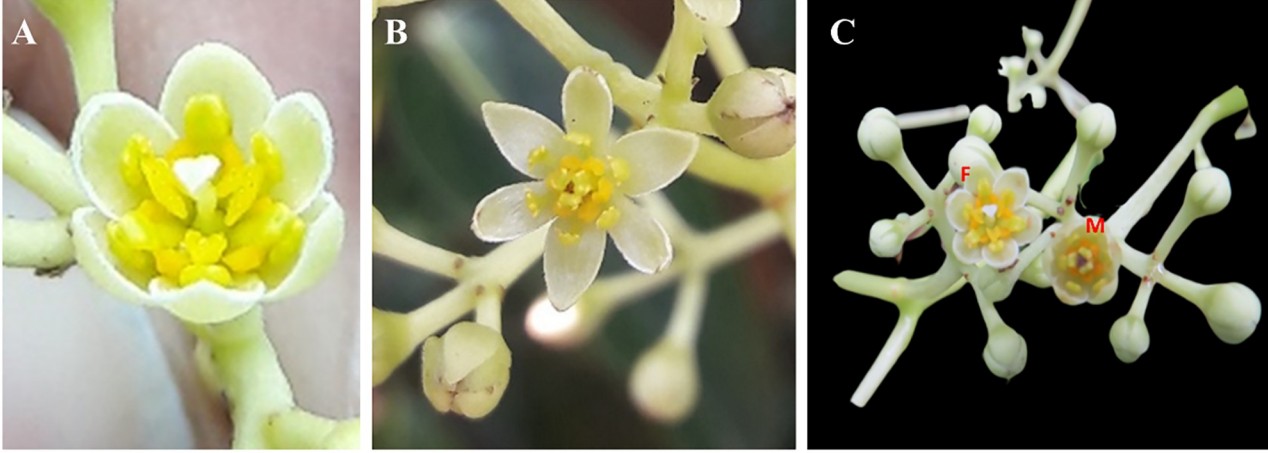

**Fig 2. Floral morphology of female and male flowers.** (A) fully opened female flower (B) fully opened male flower (C) female and the male stages opened in the same tree which can result in close-pollination.

## Flower development of type A and type B cultivars

In the *Sri Gemunu* cultivar, flowers open for the first time in early or mid-morning, remain open and the pistil is receptive until about noon. Then the flower closes and remains closed until about noon on the second day. When those reopen, begin shedding pollen with the pistil no longer receptive. Finally, those close permanently on the second day evening. The complete opening and closing cycle of *Sri Gemunu* spans about 36 hours. On the same tree, many flowers open for the first time in the same morning and then follow the same behaviour pattern synchronously hour after hour for their 2-day existence. The flowers of the *Sri Wijaya* cultivar function analogously but with transposed timing. The opening cycle of *Sri Gemunu* spans about 24 hours. The difference in cycle time reflects the relative length of the closed period between openings. Anther dehiscence occurs after 30 minutes to 1 hour from the second flower opening. The surface of the stigma turns brown and shrivelled by the time pollen is released (Fig 3).

**Female stage.** The female stage flower is half open with a white stigma that stands erect and separate, ready to receive pollen (Fig 4A, 4C and 4D) though the pollen is not released from the closed pollen sacs of that flower (Fig 4F). The stigma was tiny, papillose, and slightly capitate; the single uni-ovulate carpel was associated for up to 80% of the ovary and plicate above this and along with the elongate style (Fig 4G and 4H). Female cinnamon flowers have receptive stigmata, silky white (Colour code: 1 A 1; colour values according to Kornerup and Wanscher Handbook [23]) (Fig 4H). The gynoecium is produced by a unilocular pistil and contains one ovule (Fig 5H). Stamens are free, 9+3 in four whorls in three each (Fig 4A and 4B). The anthers are with unopened pollen sacs at the end of the stamens (Fig 4J). It has all 12 stamens bent at almost a 90-degree angle to the central erect pistil (Figs 1A and 4D). Nectar is secreted from the three staminodes located at the base of the third whorl of stamens (Fig 4E, 4F and 4I).

**Male stage.** The flower opens as a functional male after closing overnight following the female opening. The male flowers were fully open when the stigma turned brown or black and

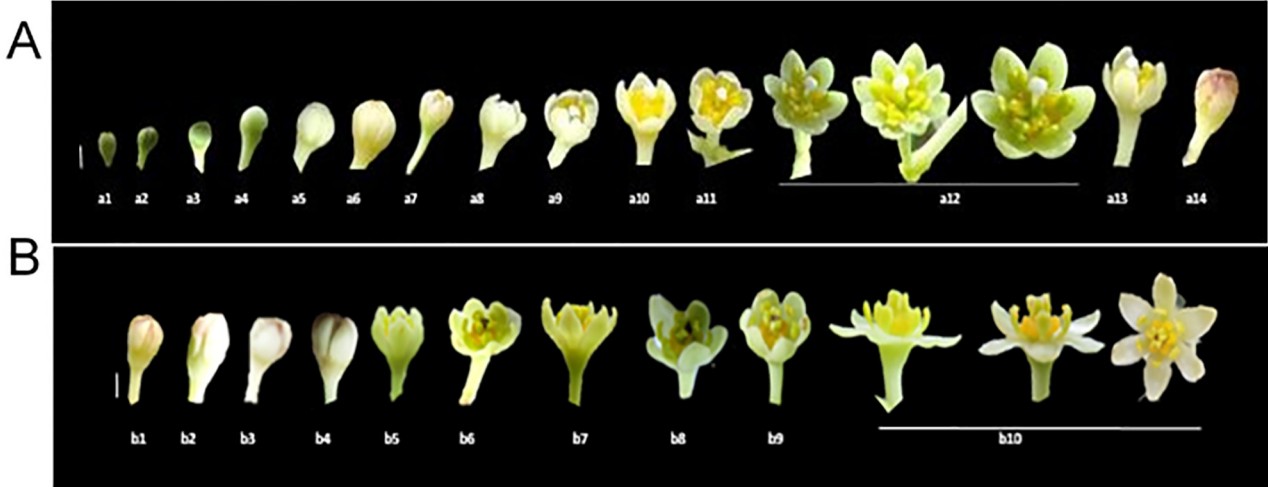

**Fig 3. The sequential floral development of cinnamon; illustrating the development of newly emerged flower bud to fully developed female flower and subsequently to male flower.** (A) a1-a14 Changes of flowering in the female flower during the first phase. a1-a4 immature flower bud with light green surface; a5 small-sized bud with light green surface; a6- medium-sized mature bud with yellow surface; a7-a9- Initiating bloom; a10-a11- half blooming stage; a12- fully blooming stage; a13-a14 wither stage (a bar- 1 mm). (B) b1-b10 Changes of flowering in the male flower during the second phase. b1-b4 initial flower bud with pale colour; b5-b7- initiating bloom; b8-b9- half blooming stage; b10- fully blooming stage (b bar– 2 mm).

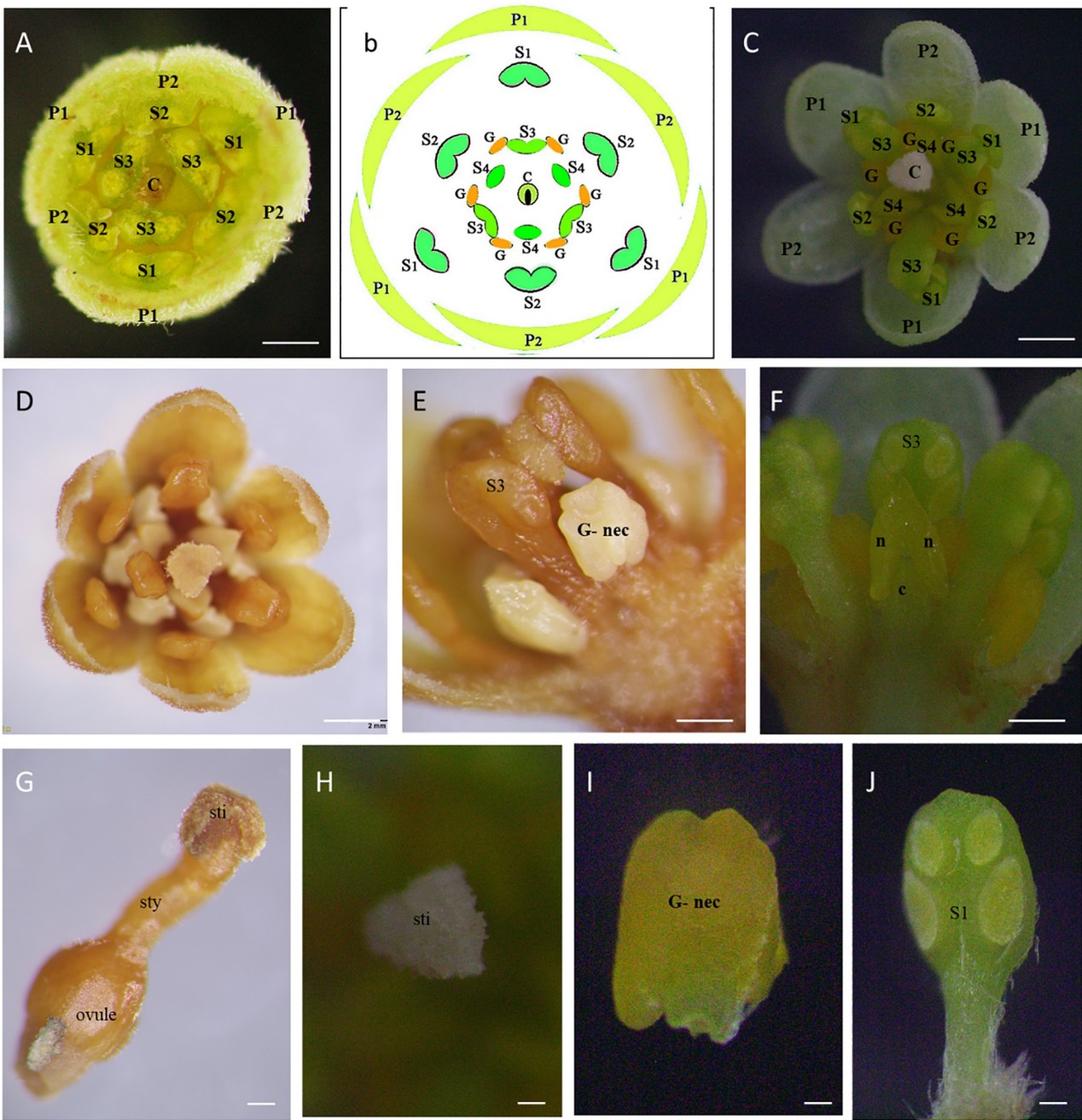

**Fig 4. Flower and floral organs of *Cinnamomum verum* functional female flower.** (A) Transverse section of flowers (B) Schematic representation of assumed floral organization in *Cinnamomum verum* flower (C,D) Flower at the female stage, apical view, dissecting microscopy (E) Stamen of the third whorl during the female stage, with a pair of paired staminal appendages, abaxial view (F) Slightly aberrant staminode (fourth androecial whorl) before anthesis, with lateral nectary tissue separated by a constriction, reminiscent of pollen sacs, apparently separated by a prominent mid-portion connective, adaxial view (G) Pistil, ovary in longitudinal-section, showing single, apical ovule (H) Dorsal ventral view of female receptive stigma, milk-white triangular shaped (I) adaxial view of nectariferous structure without anther (J) adaxial view of the unopened stamen (S1) Abbreviations. C, carpel; G, gland; P1, the outer whorl of tepals; P2, the inner whorl of tepals; S1, the first/outermost whorl of stamens; S2, the second whorl of stamens; S3, the third androecial whorl including either fertile stamens or staminodes; S4, the fourth androecial whorl including staminodes.

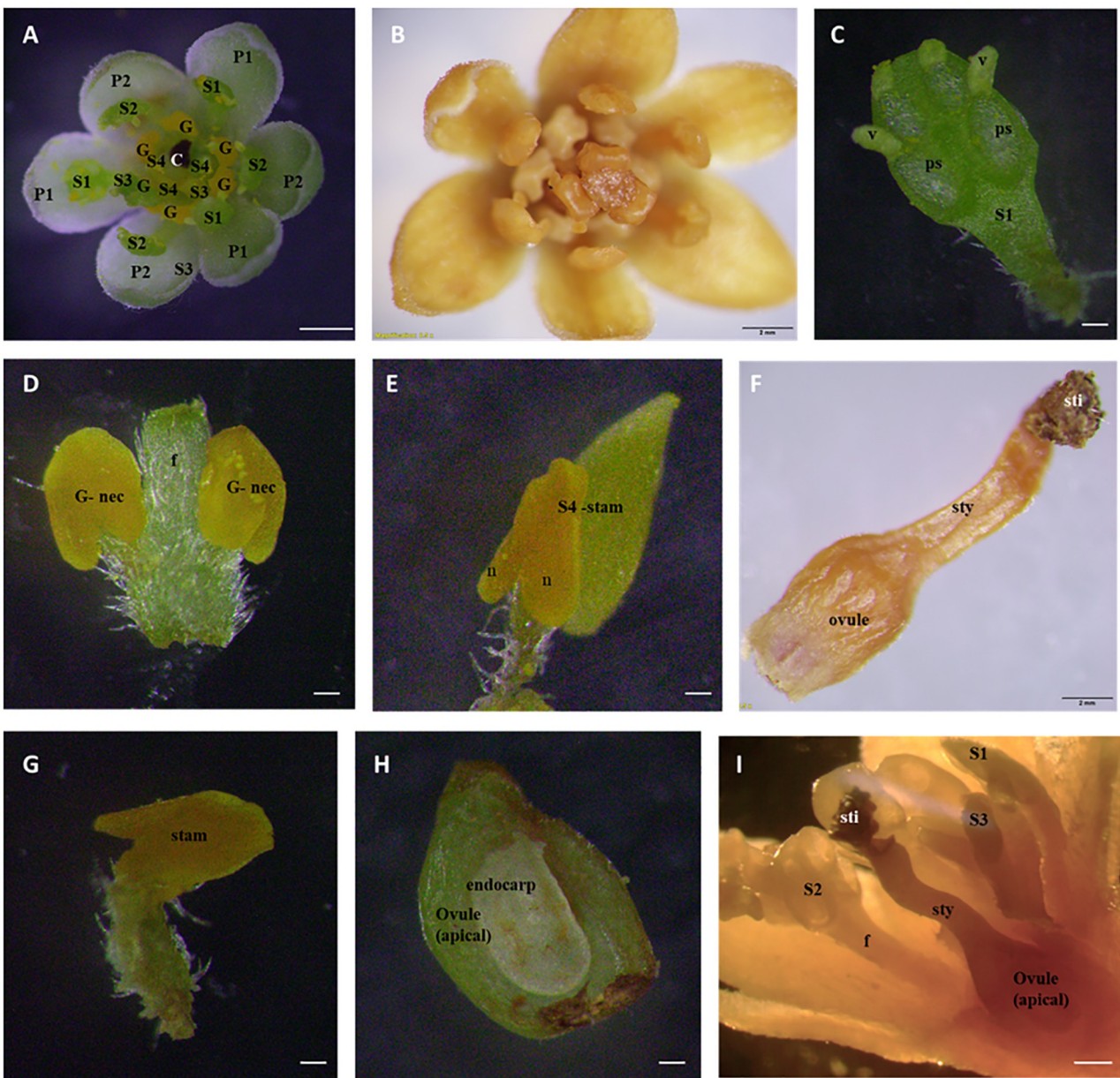

**Fig 5. Flower and floral organs of *Cinnamomum verum* functional male flower.** (A,B) Transverse section of male flower, at anthesis (C) First whorl stamen at anthers with open stoma flaps, the base has no trace of additional structures, adaxial view (D) Close up of pair of appendages at the base of a third whorl stamen, abaxial view (E) Aberrant inner tepal at anthesis, with yellow, putatively nectariferous structure at the base, with slightly aberrant staminode at anthesis, with lateral nectary tissue separated by a constriction (F) pistil, darkened stigma pollinated, with style and the ovary (G) the fourth whorl stamen lateral view, (H) ovary in longitudinal-section, showing single, apical ovule (I) lateral view of pollinated flowers with opened stamens in the three whorls and stigma darkened with fibrous-like structures.

wilted at the center (Figs 2B, 5A, 5B, 5F and 5I). The androecium has four trimerous whorls (floral whorls 3–6), with the outermost whorl alternating with the inner tepals; three outer whorls of fertile stamens and an inner sterile whorl of staminodes; and three outer whorls of fertile stamens, and an inner sterile whorl of staminodes (floral whorl 6, androecial whorl 4) (Fig 5B). Mature stamens are longer upright, prominent, and sticky. Inner three stand erect in the middle around the pistil, and the other six stand at an angle of 50-60° closer to the pistil

(Fig 5A and 5B). Between the internal and exterior stamens, those form a ring at the base, identified as the floral nectary (Fig 5D, 5E and 5G). There is no obstruction to nectar entry other than the gap between internal and external stamens (Fig 5D and 5E). The anther dehiscence occurs after 1–2 hours from the second opening of the flower. The surface of the stigma turns brown normally and shrivelled by the time pollen is released (Fig 5I).

The stamens of the two outer whorls (androecial 1,2) were essentially identical and had a long, hairy filament as well as a cylindrical or box-shaped anther (S1 Fig). Each anther has four half-thecae with elliptical valves that only dehisced in male individuals' flowers that opened introrsely. Only those individuals had pollen in their anther thecae, which was visible on the former inner side of the constructed valves (Fig 5C). The extrorse anthers and the presence of a pair of basal staminal appendages distinguished the stamens of the following inner whorl (androecial whorl 3) from those of the outer whorls. Both appendages were placed at the filament's base and had a white, hairy stalk-like filament (Fig 5J).

Accordingly, the apices of the appendages are kidney-shaped, hairy, triangular, and comparable to the apical structure of staminodes at maturity. The appendage apex's yellow bulge was more visible on the adaxial side (to the floral axis) (Fig 5F and 5E). The staminodes (androecial whorl 4) had a shorter filament and a yellow swelling apex, triangular-acute with a tip akin to an apical connective appendage, and more noticeably bulging adaxially, with a larger connective on the abaxial side. Nectar was secreted by staminodes and paired staminal appendages.

The open *Cinnammomum* male and female flowers were compared. The male flowers were considerably larger (Table 1). The male stage flowers were fully opened, and the female stage flowers were concave-shaped. Significant differences were in the median flower diameter

**Table 1. Floral characteristics of *Cinnamomum verum* Sri Wijaya and Sri Gemunu.**

| Floral characteristics | | Sri Gemunu | | Sri Wijaya | |
|---|---|---|---|---|---|
| | | **Female** | **Male** | **Female** | **Male** |
| Flower length (mm) | Min-max | 6.8–10.3 | 7.3–11.1 | 6.75–10.6 | 7.35–11.45 |
| | Ave±SD | 8.58±1.27[b] | 9.08±1.31 | 8.80±1.25[b] | 9.10±1.30[b] |
| Flower diameter (mm) | Min-max | 5.2–6.6 | 5.3–6.79 | 5.2–6.5 | 5.3–6.8 |
| | Ave±SD | 6.09±0.39[c] | 6.22±0.40 | 6.11±0.37[c] | 6.23±0.39[c] |
| Pedicle length (mm) | Min-max | 0.73–1.2 | 0.73–1.2 | 0.75–1.5 | 0.75–1.5 |
| | Ave±SD | 0.93±0.13[gh] | 0.93±0.13 | 0.98±0.21[g] | 0.98±0.21[h] |
| Inflorescence length (mm) | Min-max | 3.5–17.4 | 3.5–17.4 | 3.8–17.4 | 3.8–17.4 |
| | Ave±SD | 13.43±4.47[a] | 13.43±4.47 | 13.19±4.15[a] | 13.19±4.15[a] |
| Stigma diameter (mm) | Min-max | 0.25–0.62 | 0.24–0.59 | 0.26–0.58 | 0.28–0.60 |
| | Ave±SD | 0.45±0.13[h] | 0.42±0.14 | 0.45±0.09[h] | 0.48±0.10[g] |
| Style length (mm) | Min-max | 0.91–1.8 | 0.78–1.5 | 0.9–1.75 | 0.80–1.49 |
| | Ave±SD | 1.35±0.3[fg] | 1.13±0.23 | 1.38±0.26[f] | 1.15±0.19[g] |
| Petal length (mm) | Min-max | 2.89–3.5 | 2.89–3.5 | 2.75–3.5 | 2.75–3.5 |
| | Ave±SD | 3.13±0.18[d] | 3.13±0.18 | 3.14±0.21[d] | 3.14±0.21[d] |
| Petal width (mm) | Min-max | 1.2–1.64 | 1.2–1.64 | 1.35–1.63 | 1.35–1.63 |
| | Ave±SD | 1.5±0.11[f] | 1.5±0.11 | 1.49±0.09[f] | 1.49±0.09[f] |
| First and second whorl anther length (mm) | Min-max | 1.8–2.23 | 2–2.55 | 1.92–2.25 | 2.03–2.6 |
| | Ave±SD | 2.1±0.12[e] | 2.4±0.15 | 2.11±0.15[e] | 2.41±0.14[e] |
| Flowers per inflorescence | Min-Max | 28–158 | | 18–216 | |
| | | 81.5±45.61 | | 101.52±70.38 | |

Values are with means ± SD of all the female and male stage flowers in Sri Gemunu and Sri Wijaya (n) = 10 replicates, Means denoted by the different letters within the same column are significantly different at *P<0.05*

(male: 6.22 mm in *Sri Gemunu* 6.23 mm in *Sri Wijaya* vs. female: 6.09 mm in *Sri Gemunu* and 6.11 mm in *Sri Wijaya*), the mean stamen length (male: 2.4 mm vs. female: 2.1 mm), and the mean pistil height (male: 1.15 mm vs. female). Even though the female pistils and stigmata were higher than the male pistils, those were much taller than the surrounding anthers.

## Flowering period, floral morph proportion and flower life span

The flowering of *C. verum* occurs mainly from the end of December to the middle of February. Flowering amplitude was the highest in the mass flowering period at the beginning of February (Fig 6A and 6B). The total duration of the flowering period in a population was about one month. For an average flower, it takes 13–15 days for development from the day of initiation. The opening of the flowers occurred in two stages; the first opening was the female stage, and the second opening was the male stage. It looks receptive at the first stage after the flower opened, and there was no anther dehiscence (Fig 2A). The flowers closed and opened again the next day, i.e., the second stage when the anther dehisces in the second time opening of the flower (Fig 2B).

## Floral behavior: Overlapping period

As a result of mass flowering, under ideal temperatures (maximum 32˚C, minimum 27˚C) (S1 and S2 Figs) there was some overlap with other flowers providing a small window of opportunity for close pollination. The maximum overlapping period was seen on the day having a temperature 29˚C-32˚C in both the cultivars (S2 and S3 Figs). A long overlapping period of female and male flowers was observed in *Sri Gemunu* from around 10.00 am– 11.45 am, while in *Sri Wijaya* female and male flowers overlapped between 12.00 am-12.35 am (Fig 7A). Based on our results, the optimum temperature and humidity for overlapping during mass flowering were 29˚C-30˚C and >70%, respectively (S2 and S3 Figs). It provides a small window of opportunity for close pollination. We found an overlap of 45 min to 90 min in *Sri Gemunu*, while it was 45 minutes in *Sri Wijaya* (Fig 7B).

During low temperature (maximum 25˚C, minimum 22 ˚C) and high humidity (81.2%) (S2 and S3 Figs) the flower opening, and closing were delayed and extended (Fig 7B). However, during cooler conditions, flower opening was delayed and extended. During cooler days,

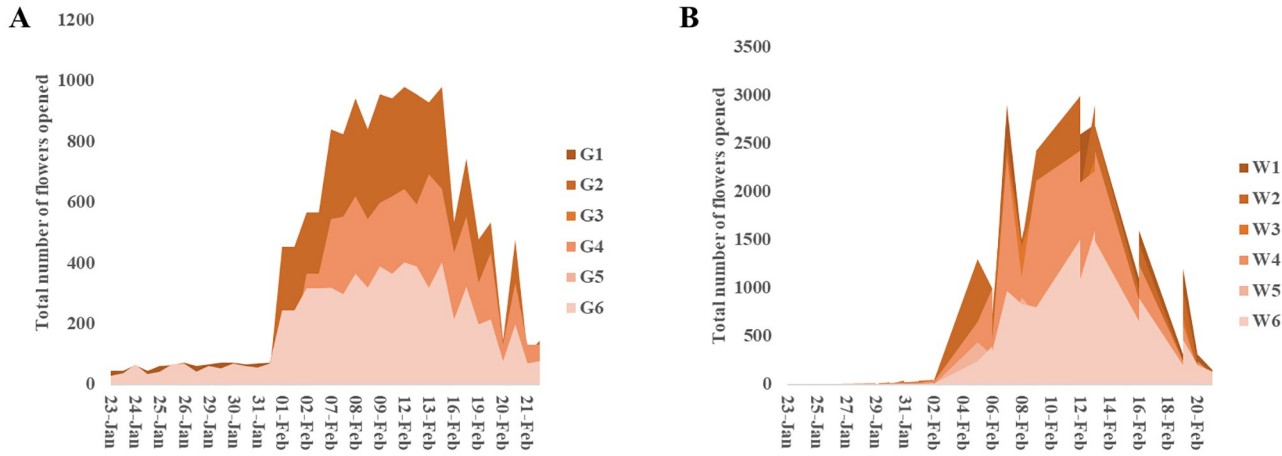

**Fig 6. Progression of flowering in (A) *Sri Gemunu* and (B) *Sri Wijaya* at Delpitiya Sub research station in 2019.**

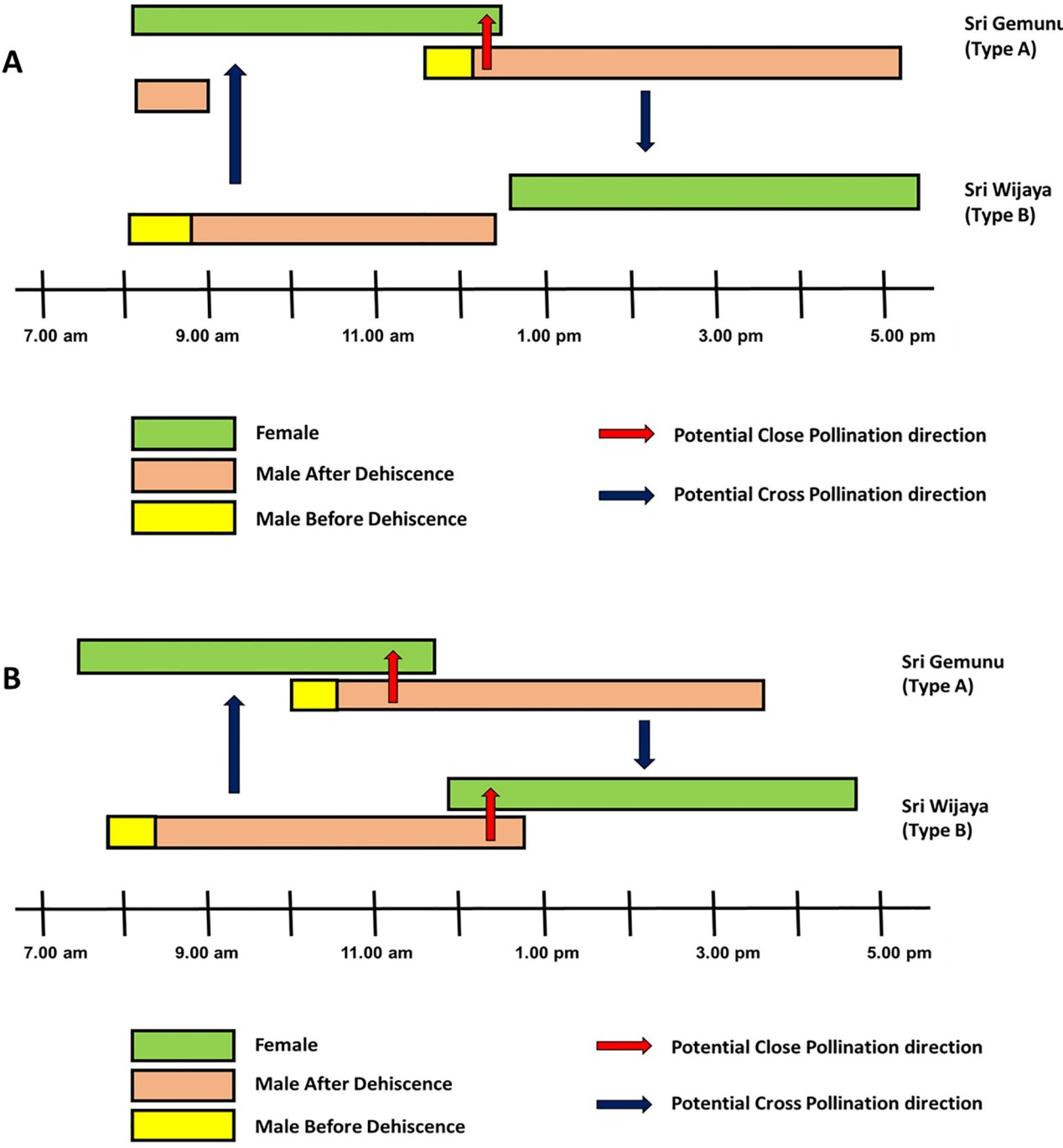

**Fig 7. Floral behaviour of male and female flowers in the two verities under different environmental conditions.** (A) Flower opening sequence of *Cinnamomum verum (Sri Lankan)* under ideal temperature when all open flowers in a block are considered. The arrows indicate the potential pollen flow (B) Flower opening sequence in *Cinnamomum verum (Sri Lankan)* trees under cold weather conditions (temperature <25 ˚C) experienced in Delpitiya, Sri Lanka-when all open flowers in a block are considered. The arrows indicate the potential pollen flow.

the opening of the *Sri Gemunu* male phase got delayed and then remained open till late evening (Fig 7B). A similar trend was observed in the female stage of *Sri Wijaya* (Fig 7B). Therefore, overlapping male and female stages on the same plant did not happen during the cooler weather.

**Table 2. Correlation of *Sri Gemunu* and *Sri Wijaya* floral overlapping with climatic factors.**

|  | Environment Factor | Coefficient | t | DF | P value |
|---|---|---|---|---|---|
| *Sri Gemunu* | Mean relative humidity (%) | -0.531 | -13 | 430 | <0.0001* |
|  | Maximum daily temperatures (°C) | 0.484 | 11.47 | 430 | <0.0001* |
| *Sri Wijaya* | Mean relative humidity (%) | -0.535 | -9.03 | 214 | <0.0001* |
|  | Maximum daily temperatures (°C) | 0.477 | 7.95 | 214 | <0.0001* |

*Significant at p<0.001

Spearman's rank correlation between the number of individuals in overlapping phases in both *Sri Gemunu* and *Sri Wijaya* is depicted in Table 2. There is a positive correlation between the flower overlapping with temperature in *Sri Gemunu* (r = 0.484, p = <0.0001) and *Sri Wijaya* (r = 0.477, p = <0.0001). It is negatively correlated to mean relative humidity in *Sri Gemunu* (r = -0.531, p = <0.0001) and *Sri Wijaya* (r = -0.535, p = <0.0001).

Of the tested climatic parameters, average daily mean temperature showed a significant correlation to the peak of male and female stage flower overlapping in the same tree (Table 3). The peak overlapping was related to the average maximum temperature, but not the average minimum temperature. This was associated with the decrease in the relative humidity in the air. The linear regression between the overlapping, with average mean temperature and average minimum humidity, could explain much of the observed variation in the rate of overlapping in the two cultivars (Fig 8).

Table 4 shows the synchrony of phenological events of the two cultivars *Sri Wijaya* and *Sri Gemunu*. The highest synchrony ratio indicates a greater coincidence of the phase among individuals. The overall inter-population synchrony ratio for flower overlapping between male and female flowers during the overlapping period is higher during the peak season. The overall synchrony ratio during peak season/mass flowering during January and February for *Sri Gemunu* and *Sri Wijaya* are 0.873± 0.040 and 0.854±0.026, respectively (Table 4).

## Receptivity of the stigma

The stigma supports pollen hydration, germination, and initial pollen tube growth, and the length of stigmatic receptivity plays an important role in the effective pollination period and subsequent fruit set [24, 25]. The stigma receptivity continued up to the male stage since the female flower opening. The highest stigmatic receptivity was observed between 8.30 am to 10.30 am in *Sri Gemunu*, while for *Sri Wijaya* it was around 1.00 pm to 4.00 pm (Fig 9, Table 5). These results suggest that the *Cinnamomum* stigma maintains its capacity to support pollen adhesion and pollen germination during the length of the female phase. Thus, the pollen could be applied to the stigma along the female stage during hand-pollination although the stigma seems to provide higher adhesion during the middle part of this period.

**Table 3. Linear regression analysis of peak overlapping flowering with climatic factors.**

|  | *Sri Gemunu* | | *Sri Wijaya* | |
|---|---|---|---|---|
| Parameter | Temp | Humidity | Temp | Humidity |
| Intercept | 76.9** | (-)16.53** | 136.2** | (-)20.84** |
| SE coef | 28 | 4.58 | 15 | 3.34 |
| Adjusted R-sq | 0.27 | 0.44 | 0.86 | 0.76 |

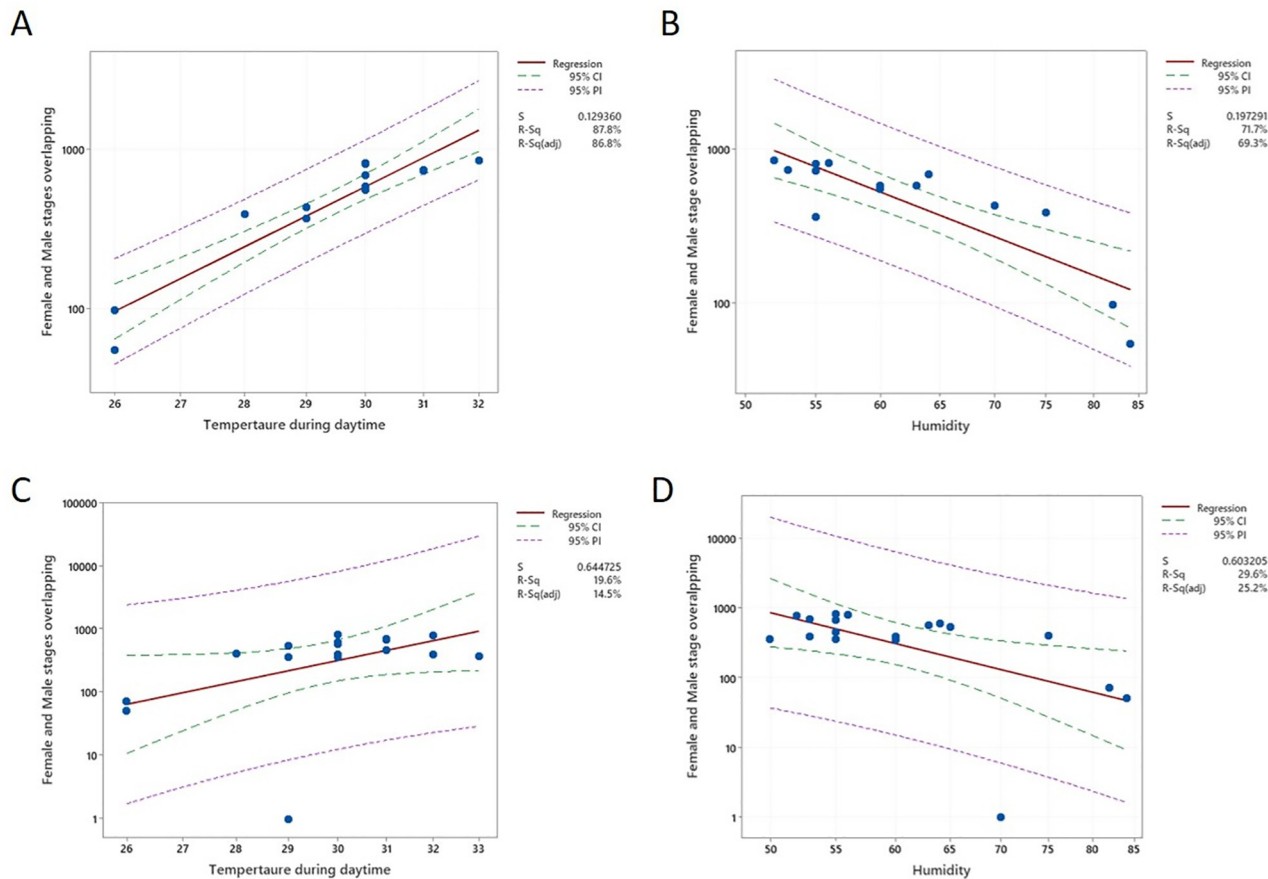

**Fig 8. Influence of average daily mean temperature and humidity on rate of floral overlapping during peak flowering season.** The fitted line represents predicted values for the rate of progress from January to peak flowering date according to linear regression. (A) Graphical representation impact of temperature on *Sri Gemunu* female and male stage flower overlapping in peak season February. (B) Graphical representation impact of humidity on *Sri Gemunu* female and male stage flower overlapping in peak season February. (C) Graphical representation impact of temperature on *Sri Wijaya* female and male stage flower overlapping in peak season February. (D) Graphical representation impact of humidity on *Sri Wijaya* female and male stage flower overlapping in peak season February.

## Pollen morphology and pollination biology

The SEM images describe the morphology of the pollen grains of two Cinnamon cultivars (Fig 10), and the quantitative and qualitative features are summarized (Table 6). The pollen grains of the investigated *C. verum* cultivars are spheroidal in shape, apolar and inaperturate. As expected, both cultivars have the same shape pollens (Fig 10A and 10E). The exine ornamentation is spinulate with a spine having a conspicuous cushion base. This cushion base is prominent and ornamented with balls-like structures. According to Erdtman's pollen size

**Table 4. Synchrony indices for phenological events of *Sri Gemunu* and *Sri Wijaya* within peak and off-peak season.**

| Variety | Season | SI | temperature | humidity |
|---|---|---|---|---|
| *Sri Gemunu* | Peak | 0.873± 0.040[a] | 30.83 | 59.3 |
| | Off peak | 0.687±0.049[b] | 26.83 | 76.7 |
| *Sri Wijaya* | Peak | 0.854±0.026[a] | 30.83 | 59.3 |
| | Off peak | 0.6239±0.031[b] | 26.83 | 76.7 |

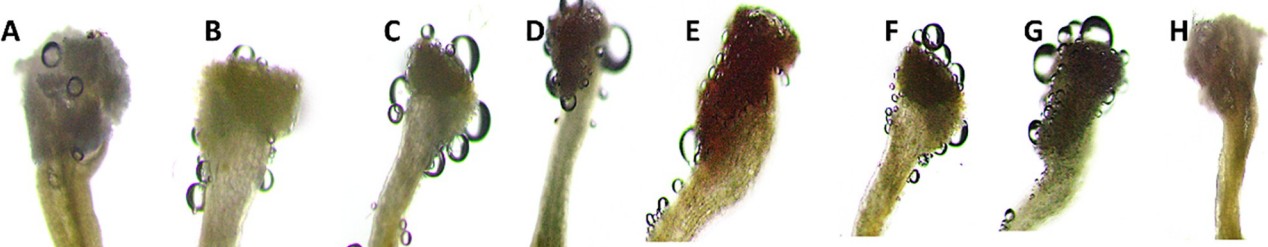

**Fig 9. Evaluation of the stigmatic receptivity of *Cinnamomum verum* with hydrogen peroxide.** (d-g); A) No reaction in pre-anthesis; B) Strong positive reaction (+) in anthesis; C) Very strong positive reaction (++) in after 2 hours; Very strong positive reaction (++) in after 2 hours D) Weak positive reaction (+) in pre-anthesis; E) Strong positive reaction in anthesis; F) Very strong positive reaction in anthesis; G) Very strong positive reaction in post-anthesis.

**Table 5. Stigma receptivity in *C. verum* of the two cultivars *Sri Gemunu* (SG) and *Sri Wijaya* (SW) evaluated by hydrogen peroxide at different times related to anthesis.**

| Variety | 7.30 am (pre anthesis) | 8.00 am (pre anthesis) | 9.00 am (pre anthesis) | 10.00 am (pre anthesis) | 11.00 am (pre anthesis) | 12.00 noon (anthesis) | 1.00 pm (anthesis) | 5.00 om (post anthesis) |
|---|---|---|---|---|---|---|---|---|
| SG | + | ++++ | ++++ | ++++ | +++ | +++ | ++ | - |
| Variety | 7.30 am (anthesis) | 11.00 am (anthesis) | 12.00 pm (pre anthesis) | 1.00 am (pre anthesis) | 2.00 am (pre anthesis) | 3.00 noon (pre anthesis) | 4.00 pm (pre anthesis) | 5.00 om (post anthesis) |
| SW | + | + | + | ++ | +++ | ++++ | ++++ | - |

(−) no reaction; (+) weak positive reaction; (++) strong positive reaction; (+++) very strong positive reaction. Method adapted from Dafni and Maués (1992) [22]

classification [26], most of the investigated pollen grains were median with the polar axis (P) of corpus ranging from 22–24 µm. *Sri Gemunu* pollen is larger than that of *Sri Wijaya* in size. The spine length ranges from 0.8-to 0.9 µm and of the interspinal space is higher in *Sri Gemunu* than the *Sri Wijaya* (Fig 10B and 10F). The spinules of *Sri Wijaya* pollen are shorter

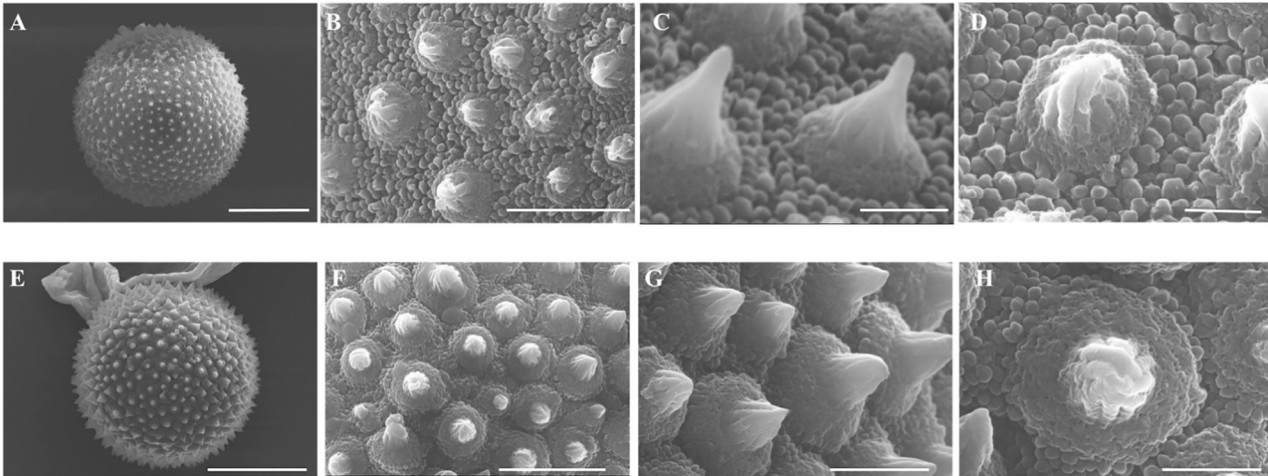

**Fig 10. Scanning electron microscopy (SEM) images of pollen grains of *Cinnamomum verum (Sri Lankan)*; Sri Gemunu and Sri Wijaya.** (A,B,C,D) *Cinnamomum verum (Sri Lankan)* Sri Gemunu Type A (A) Polar view of pollen grains (B) detail of spines and exine structure /Exine ornamentation (C) Dorsal view of an enlarged spine, tectum is highly perforated (D) Ventral view of the enlarged spine, striated conical-shaped spine on a cushion base (E,F,G,H) *Cinnamomum verum (Sri Lankan)* Sri Gemunu Type A (E) Polar view of pollen grains (F) detail of spines and exine structure /Exine ornamentation, tectum is less perforated (G) Ventral view of a Striated Conical linear spine on a cushion base (H) Dorsoventral view of a spine.

**Table 6. Quantitative and qualitative pollen morphological characters of *Cinnamomum*.**

| | Quantitative | | | | | |
|---|---|---|---|---|---|---|
| | Polar length (P) μm | Length of the spinule μm | Length of the spinule with base | Interspinular Distance μm | Diameter of the spinule base | |
| *Sri Gemunu* | 24.36±0.123 | 0.86±0.082 | 1.11±0.113 | 0.48±0.103 | 0.86±0.082 | |
| *Sri Wijaya* | 22.22±0.123 | 0.89±0.012 | 1.48±0.018 | 0.80±0.100 | 1.60±0.007 | |
| | Qualitative | | | | | |
| | shape class of pollen | Tip of spines | Ornamentation of spines | Types of granules | Nature of granules | Nature of cushion base |
| *Sri Gemunu* | spheroidal | Acute | Striated | Monomorphic | Prominent | Prominent |
| *Sri Wijaya* | spheroidal | Acuminate-acute | Striated | Monomorphic | Prominent | Prominent |

than that of the *Sri Gemunu. T*he spinule tip of the *Sri Gemunu* pollen is pointed, conical shaped, and striated (Fig 10C), while the *Sri Wijaya* are acute and blunt, conical linear, and striated (Fig 10G). Moreover, the tectum is highly perforated in the *Sri Gemunu*, while it is less in the *Sri Wijaya* (Fig 10D and 10H).

Interestingly, the *Sri Gemunu* and *Sri Wijaya* had a significant difference between each other in floral level. Therefore, we looked at whether there is a correlation between the quantitative characteristics for the species delimitation (Fig 11). However. there was no significant correlation between the quantitative morphological traits of flower and pollen.

## Floral visitor observations

According to the field observations, flowers of *C. verum* are assumed to be pollinated by insects including mainly Hymenoptera, Diptera Syrphidae, and Coleoptera. Namely, honeybees (*Apis mellifera*, *Apis cerana*), Hoverfly (*Dideopsis aegrota*), Wasps (*Delta dimidiatipenne*), bee (*Trigona iridipennis*), Butterfly (*Ypthima ceylonica*), and *Formicidae* sp (*Oecophylla smaragdina*), flies (Housefly *Lucilia cuprina*) and mosquitoes (Fig 12). They forage daily during day hours from 0800-1800h collecting pollen and nectar. When visiting flowers, bees grasped petals by using their feet to fix the body, generally staying on a single flower for about 10–30 seconds and then flying to another flower.

The average duration of the visit made by hoverflies is 5–25 sec; *Wasps* with 1 minute between 10.00 and 18.00h, *Apis mellifera* visits for 8–25 sec between 09.00 to 13.00h. Butterfly visits for 2 to 5 sec from 09.00 to 12.00h and the duration of ants' visit is highest between 08.00 to 18:00h. The honeybees and the hoverflies are the dominant pollinators. Our observations suggest that excluding mosquitoes and thrips that may act as nectar thieves, other floral visitors were effective pollinators that could perform pollination while they were collecting pollen or sucking nectar.

## Reproductive output under controlled hand pollination

The percentage of fruit set in the three plants isolated is given in Table 7.

The fruiting rate was around 4% (Fig 5) in both controlled insect-pollinated conditions and around 8% (Fig 5) in hand-pollinated conditions. However, no fruiting was observed in the isolated plant that was not pollinated.

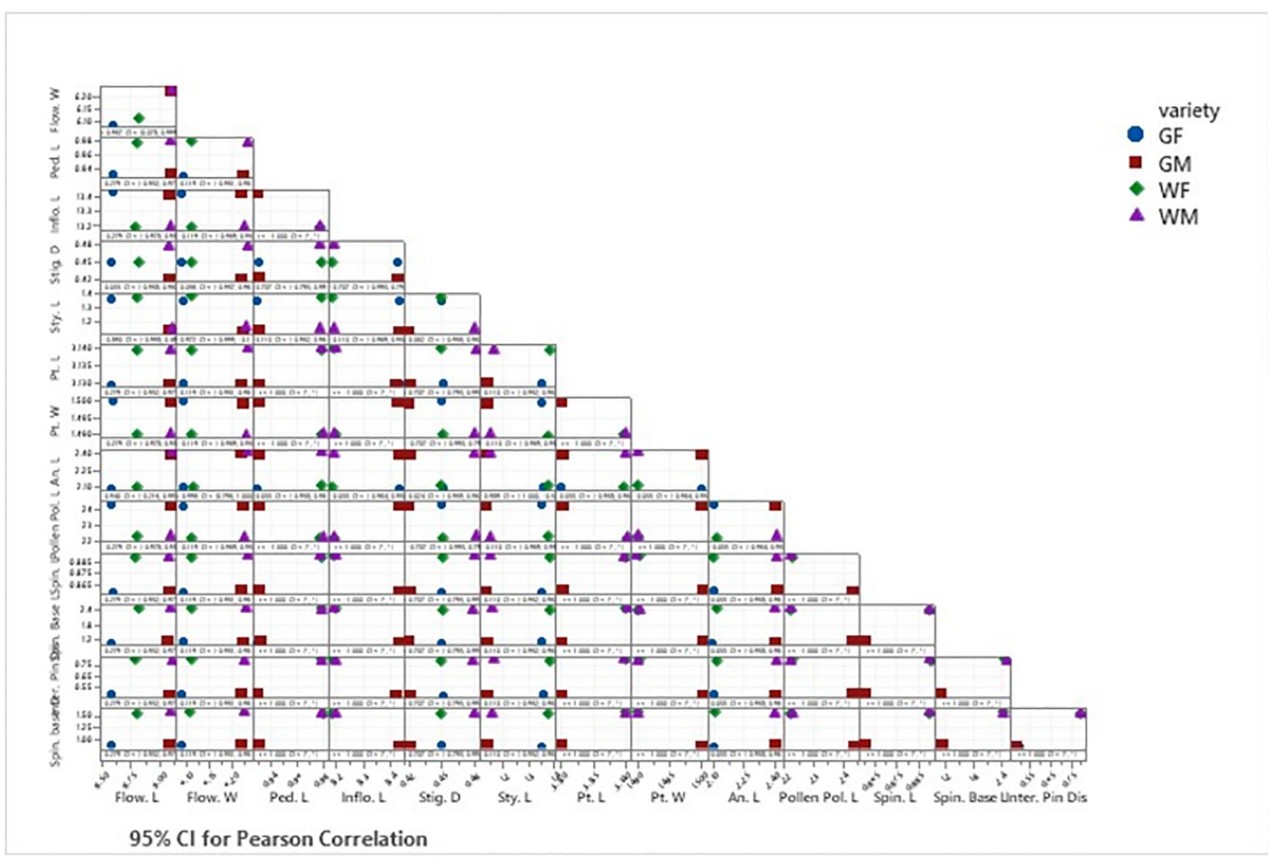

**Fig 11. Correlation between morphological characters of flower and pollen.** Correlation matrix through Pearson correlation coefficient, *Significant at p<0.05.

### Histological evidence for self-incompatibility

Initial microscopic observation suggested that in close-pollinated flowers, crossed by hand, the pollen was reaching the stigmatic surface of the pistil, belonging to the same tree, and started to germinate indicating their viability. For the enhanced resolution of pollen-stigma interaction up to the pre-fertilization stage, a further microscopic study was conducted with the pistils collected at different stages of the floral cycle in the female stage. Subsequent pollen tube growth showed a differential pattern between cross-pollinated and close-pollinated flowers to cover the stylar canal during the first 24 hours. The self-pollinated pistils developed cellulosic walls at and around nucleus cells surrounding the megaspore. This callose growth progressively increased with time (Fig 13E–13G), restricting reaching the ovule (Fig 13H).

During the pre-receptive period, the stigma was slightly enlarged (Fig 13A), papillae did not expand completely, and some remained relatively wrinkled. During the receptive period from around 8.00 am to 11.00 am, the stigma starts secreting exudate from vesicles and secretions seen on the surface, and some papillae had periplasmic spaces where secretions may be present. During the post-receptive period, the stigma turned blackish, and the remnants of the exudate were found on some stigmata (Fig 13B).

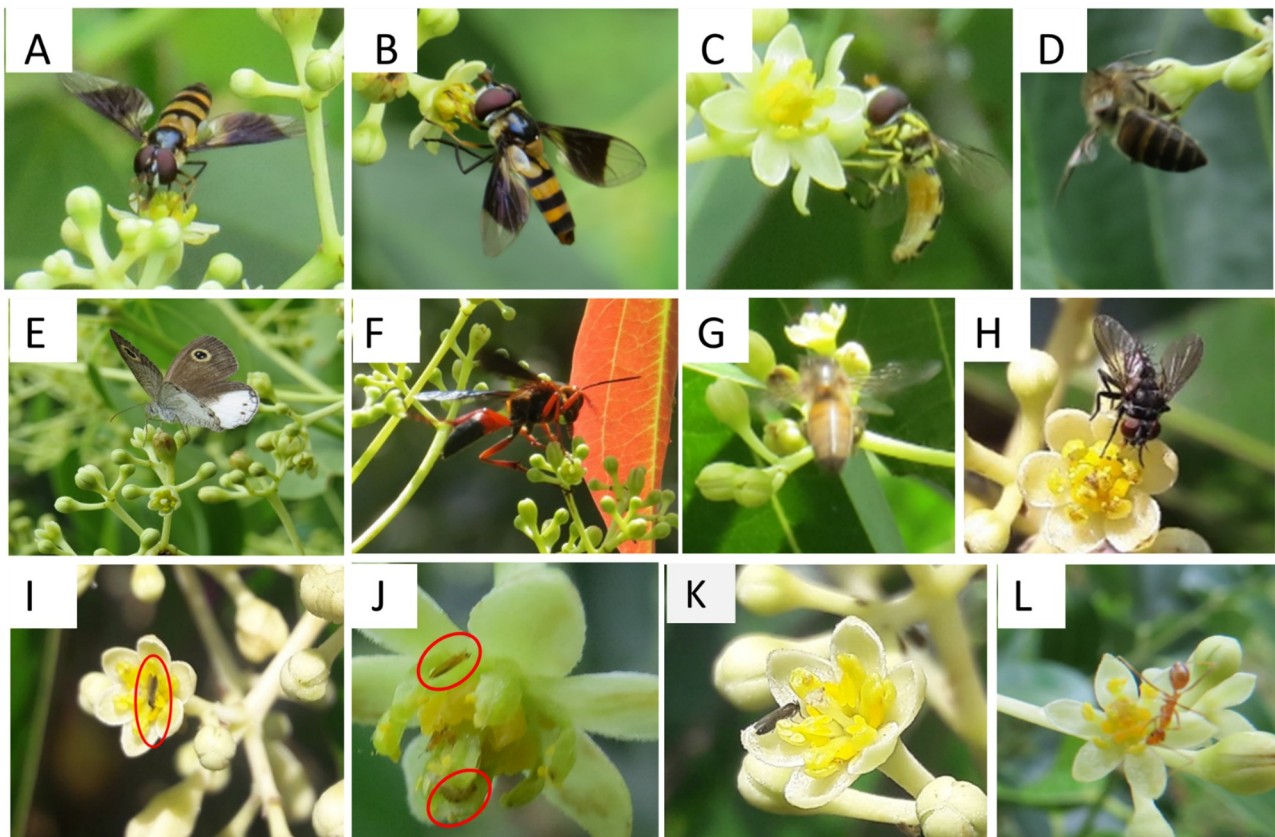

**Fig 12. Pollinator's foraging behaviors on Cinnamon flowers.** (A-B)- Hoverfly/Syrphid fly or Flower fly-*Dideopsis aegrota* (C)- Hoverfly *Syrphus ribesii* (D)- Apidae Honeybee *Apis cerana* (E) butterfly *Ypthima ceylonica* (F) Wasp Delta dimidiatipenne (G) Apidae Honeybee Apis mellifera (H) House fly *Lucilia cuprina* (I) Small flies (J)- Thrips (K) Mosquiotes (L) Formicidae Oecophylla smaragd.

## Changes of stigma surface at different flowering

The SEM observations of the stigma soon after the opening showed a prominent slightly irregular triangular-shaped stigma (Fig 14A), ending in a smooth elongated style. At the earliest stage of female opening, before the pollination, the trilobe-shaped stigma was bordered by rows of feathery appendages (Fig 14B). These feathery-like structures are the raised unicellular papillae. The papillae were closely packed and formed ridges-like structures (Fig 14C). After the stigma starts being receptive, the stigmatic papillae displayed round-shaped secretory vesicle-like structures fusing to the plasma membrane. These secretory vesicles cover the full surface for 30 minutes following compatible pollinations (Fig 14D and 14E). At this time, a thin layer of stigmatic secretion from the secretary vesicle coated the papillary surface (Fig 14F). During later stages and after pollination, the papillae became more distinct, and accumulated

**Table 7. Fruiting rate observed after controlled pollination systems.**

| Plant | Status | Total number of flowers produced | Total number of fruits produced | Percentage of fruit set |
|---|---|---|---|---|
| 01 | Insect pollinated/Plant house (higher temp/low humidity) | 46 | 2 | 4.32% |
| 02 | Hand pollinated | 148 | 12 | 8.01% |
| 03 | Isolated plant in an open area | 538 | 0 | 0% |

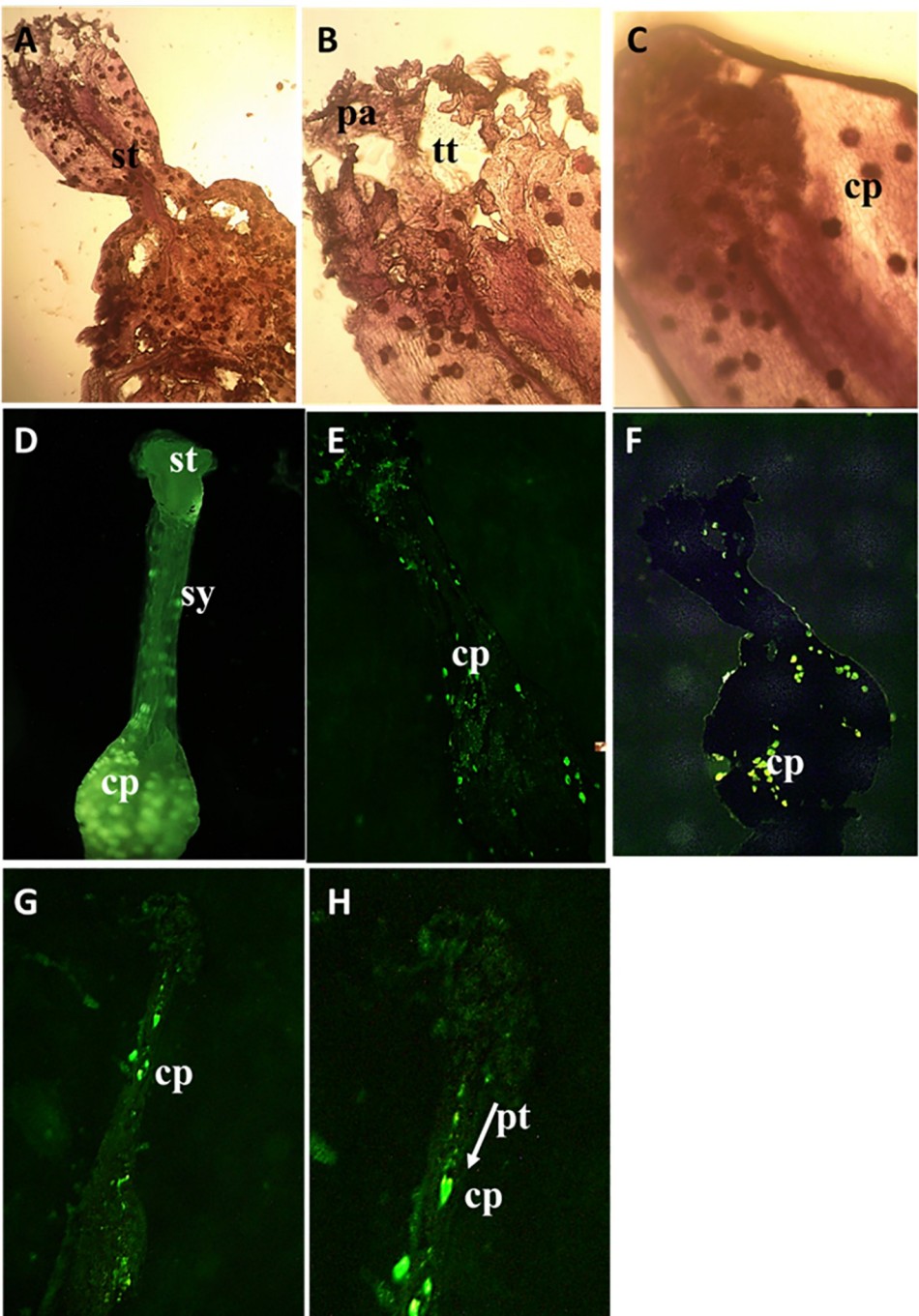

**Fig 13. Pollen tube growth analysis in incompatible and compatible assisted crosses using fluorescence microscopy.** Callose plugs (cp) have been formed so as to inhibit the pollen tube growth and fertilization (A,B,C) Light microscopic images showing the development of callose plugs (D) Close-pollinated stigma (sg) after 24 hours (E) Close-pollinated stigma showing callose plugs thickening on style and ovary area after 1 hour (F) Close-pollinated stigma showing callose plugs thickening on style (st) and ovary area after 5 hours. (G,H), aniline blue staining of self-pollinated pistil after 20 hours, showing pollen tubes within the transmitting tract and reaching the ovules. Callose plugs thickening has inhibited the growth of pollen tubes by deposition at the tube tip, arrow showing the development of pollen tube (pt).

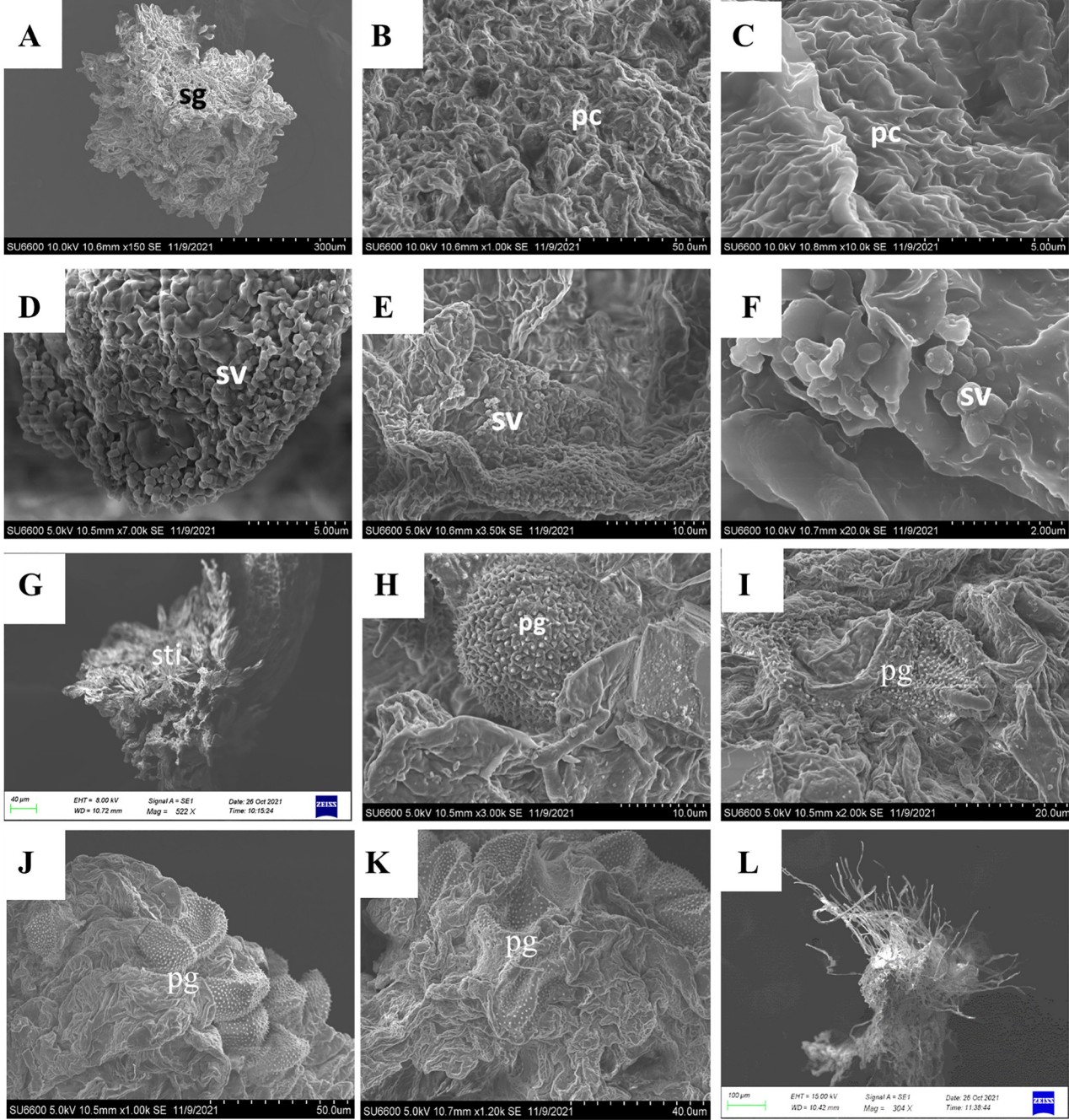

**Fig 14. Stigma morphology variation in *Sri Gemunu* during the floral cycle during close pollination.** (A) Dorsal ventral view of the fresh female stigma (sg) soon after opening (B) Stigmatic surface with elongated stigmatic papillary cells (pc) in ribs-like structures, providing higher surface area for pollen adhesion (C) Higher magnification of the papillary cell layer (D,E) Secretory vesicles (sv) appeared on the stigma after the stigma being receptive, the secretory vesicles are circular lobes-like structures, arranged all over the papillae cell layer (F) Higher magnification of secretory vesicles (G) The stigma has a central depression forming a short stigmatic cleft in the pollinated stigma, feather-like papillary cells are distinct (H) Hollen grains deposited on the female stigma, pollen merged on the stigmatic surface (I) Pollen grains deposited on the stigmatic surface shrank and shriveled after pollination (J,K) Magnified Functional male stigmatic surface with multiple pollen deposited, all those pollen grains merged into the stigmatic surface, the pollen grains are shrunk and shriveled. (L) Un-pollinated stigma in male flower, the papillae cells converted into fiber-like structures.

stigma exudate appeared at some interstitial regions at the base of the adjacent papillae. In some flowers, which appeared to be pollinated and germinated, the stigmatic surfaces collapsed forming a hollow-like thing in the middle (Fig 14G).

During stigma receptivity, the surface area of the stigma increased considerably due to the physical appendage increase in the stigmatic lobes and enhanced the exertion of globular papillae on the surface. After opening the flowers for about 2 hours (pre-anthesis), the stigma surface was densely covered with round secretory vesicles. These numerous vesicles on the stigma surface provide a receptive surface for compatible pollen grains to adhere, hydrate, germinate and grow. Within minutes of the capture of grain by the stigma, the exine-held material flows out and becomes deposited on and between the stigmatic papillae. Pollen grains were attached at the end of the papillae surface, and some were deposited on the stigma surface (Fig 14H).

After the anthesis, the stigma visually turned from milky white colour into black (Fig 2A and 2B). The pollinated stigma was visually abundant with pollen grains (Fig 14K). Most of the pollen grains were shrunk, shrivelled, and drowned into the stigmatic surface through the papillary spaces (Fig 14K and 14L). These shrunk pollen grains may be due to incompatibility. The pollen grains were not hydrated from the stigmatic surfaces and the papillae in the unpollinated stigma turned into fibre-like structures (Fig 14M). The unpollinated stigma was separated from the style (Fig 14I). There was no distinct difference between the stigmatic surfaces in *Sri Gemunu* and *Sri Wijaya* (S4 Fig), both cultivars exert similar mechanisms in pollen deposition, hydrations, and germination.

## Discussion

According to the observations, the *Cinnamomum verum* flowers are small and mostly bisexual and typically consist of seven trimerous whorls of floral organs: two perianth whorls, four androecial whorls, and one single carpel. Typically, the two perianth whorls are similar. The gynoecium in the functional female stage is considered as a single carpel and superior in most of the Lauracea [27, 28]. The stamens of the third whorl bear a pair of appendages at the base. In the functional male stage, pollen sacs open introrsely in the two outer stamen whorls, while the third inner whorl opens extensively. Anthers possess wither four pollen sacs in superimposed orientations similar to other members in Lauraceae [6, 29]. However, in male flowers of *Cinnamomum*, all stamens bear paired staminal appendages.

In addition to the morphoanatomy mechanisms exerted for cross-pollination, rewards are readily available to floral visitors such as bees, wasps, ants, flies, beetles, bugs, and butterflies. The main cinnamon pollinators are the Hymenoptera species, several small to medium size stingless bees (Meliponinae), and honeybees. *Cinnamomum sulphuratum* and avocado, having similar pollination behaviour, consider honeybees the potential pollinator [30]. Second-order pollinators are numerous species of wasps, butterflies and flies and probably beetles, too [30]. These pollinators promote cross-pollination since they have a high chance of moving among neighbouring flowers and flying between inflorescences. Flies are one of the most abundant insects detected in the pollinator profile of cinnamon. In avocado, which has a similar type of floral biology, it has been identified that the thrips (*Frankliniella* and *Thrips palmi*) play a key role in pollination [13, 31, 32]. They are common visitors of cinnamon flowers during bloom [33].

The physiological and structural originations of the floral organs also maximize cross-pollination. Assessing the morpho-anatomy and stigma receptivity of cinnamon is vital in understanding the biological process. According to our observations, *Cinnamomum* has a dry stigma, previously observed in plants having saprophytic self-incompatibility systems [34, 35].

Pollen receptivity depends on the nature of the stigmatic surface [36]. Moreover, the amount of pollen grains received by the stigma is influenced by its surface area. Accordingly, *C. verum* has a unique stigma surface in the female stage compared with the other species in the family. The receptive stigmatic dry surface is concentrated in distinct ridges with unicellular papillae, which increases pollination efficiency. This arrangement could prevent freshly deposited pollen grains from falling off the stigma and possibly be a precaution for pollen tube growth from close pollination. The numerous stigma feathers-like structures in cinnamon create a relatively larger area that increases the probability of pollen grain deposition [37, 38]. The same has been recorded in avocados [27, 39]. The secretory vesicles distinct after the stigma starts being receptive would be a critical factor affecting the pollen receptivity in the stigma, making it wet to bind the pollen.

Close-pollination barriers on the stigma surface result in the arrest of pollen germination or pollen tube entry into the stigma [40]. Normally, compatible pollens get hydrated due to the transfer of water from the stigma to the pollen through an osmotic gradient. According to our observations, pollen shrinkage and shrivelling were prominent during close pollination, possibly due to insufficient or uncontrolled hydration. However, in self-pollinated flowers, it was distinct that there was some probability of getting pollen compatible by adhesion and hydration. These pollen grains are drawn into the stigmatic surfaces through the papillary spaces.

Pollen tube growth on *Cinnamomum* is not surficial, or even immediate subcuticular, as in stigmatic papilla of *Persea americana* [39, 41] and other angiosperms with dry stigma including Brassica [40, 42] and Nymphaea [43]. It occurs deeper within the stigmatic cells. The stigmatic papillae of *Cinnamomum* are morphologically different from those of avocado [44, 45], in terms of stigmatic vesicles. Those were visually different from other dry stigmas. Thus, *C. verum* could be a useful model for comparative studies in pollen-pistil interactions in the Lauraceae family.

Moreover, the formation of callose plugs at and around the style and ovary is clear evidence of its self-incompatibility. These callose plugs form in close-pollinated pistils and may result of a self-recognition mechanism [46]. The development of such mechanical barriers in growing pollen tubes in pistils via long-distance signalling mechanisms has been extensively discussed in other families like Brassicaceae [34, 39, 47, 48]. It was evident that the size of the callose plugs increased over time, which could result in the degeneration of ovules in the close-pollinated pistil. Callose plugs observed at and around the nucleus cell/ovary may prevent nutrients from reaching the developing megaspore mother cell/embryo [47, 49].

In contrast to all these adaptations, cross-pollination and self-incompatibility systems, we observed some deviations. We achieved ~8% fruit set in *Cinnamomum* plants in a single closed environment by hand pollination, while insect pollination resulted in ~4% fruitset. This overlapping is more extensive at high temperatures and low humidity in peak seasons. Increasing the overlapping time between flowers in both male and female stages of the same plant or plants from the same variety could be a strategy to ensure close pollination. Moreover, controlling temperature and humidity extended the stigma receptivity and the overlapping period. Extended stigma receptivity assures pollination, fertilization, and reproductive success in plants [25, 35, 50, 51]. This will ensure the pollen get hydrated in the stigma and follows for successful fertilization.

Supporting the field observations, the SEM analysis showed germination of pollen grains at a considerable level during close pollination leading to successful fruiting. This is supported by a recent field study [52], in which Sri Wijaya's offspring resulted from a single open-pollinated event, 20% of individuals shared identical ISSR fingerprints with the mother plant.

Accordingly, *C. verum* has a complex reproductive mechanism. Though it is naturally adapted to cross pollination, there may be deviations in response to weather changes,

temperature and humidity. Therefore, more focused breeding strategies are needed for crop improvement to cope with increasing demand and biotic and abiotic challenges, including climate change. For example, close pollination within the tree is recommended to produce an elite seedling progeny from identified with superior mother plants or accessions. The grower will be able to enhance the seed setting from such plants under a controlled pollination environment with high temperature and low humidity.

## Supporting information

**S1 Fig. SEM images of dehisced stamens in male flower.** (A) Lateral view f Stamen Fourth whorl (S4) (B) First-whorl stamen at anthesis (with open stoma flaps); mature pollen grains at the opened pollen sac and the filament with fewer trichomes (C) Valvular dehiscence from the first whole stamen, Mature pollen grains distinct in opened pollen sac.
(DOCX)

**S2 Fig.** Progression of overlapping percentage in *Sri Gemunu* of both female and male flowers during the overlapping period in peak and off peak season, with temperature, and humidity a-f G1,G2,G3,G4,G5,G6.
(DOCX)

**S3 Fig.** Progression of overlapping percentage in *Sri Wijaya* of both female and male flowers during the overlapping period in peak and off peak season, with temperature, and humidity a-f W1,W2,W3,W4,W5,W6.
(DOCX)

**S4 Fig. Stigma morphology variation in *Sri Wijaya* during the floral cycle.** (A) Lateral view of the fresh female stigma soon after opening (B) Dorsal ventral view of the fresh female stigma (sg) soon after opening (C) Stigmatic surface with elongated stigmatic papillary cells (pc) in ribs-like structures, providing higher surface area for pollen adhesion (D,E) Secretory vesicles (sv) appeared on the stigma after the stigma is receptive, the secretory vesicles are circular lobes-like structures, arranged all over the papillae cell layer (F) Pollen grains deposited on the female stigma, pollen merged on the stigmatic surface in the female flower (G) Pollen grains submerged on the stigmatic surface for the pollen tube growth, after pollination (H) Magnified Functional male stigmatic surface with multiple pollen deposited, all those pollen grains merged into the stigmatic surface, the pollen grains are shrunk and shrivelled. (I) Pollen grain submerged on the stigmatic surface for pollen growth detected in male stigma (J) Filaments observed in the style in the male flower stigma (K) Central groove observed in the style after pollination (L) The stigma has a central depression forming a short stigmatic cleft in the pollinated stigma, feather-like papillary cells are distinct.
(DOCX)

**S1 Table. Summary of the data analysis.**
(DOCX)

## Acknowledgments

The authors acknowledge Mr. R. A. A. K. Ranawaka, Assistant Director (Research), Sub Research Station, Department of Export Agriculture, Nillambe, and the staff of Mid Country Research Station, Dalpitiya, Atabage for providing assistance. The authors would like to thank Ms P.G.L.T. Dilhani and the staff of the Agricultural Biotechnology Centre, Faculty of Agriculture, University of Peradeniya, for the support extended throughout the period. The funders

had no role in study design, data collection and analysis, decision to publish, or preparation of the manuscript.

## Author Contributions

**Conceptualization:** D. K. N. G. Pushpakumara, Pradeepa C. G. Bandaranayake.

**Data curation:** Bhagya M. Hathurusinghe.

**Formal analysis:** Bhagya M. Hathurusinghe.

**Funding acquisition:** Pradeepa C. G. Bandaranayake.

**Investigation:** Pradeepa C. G. Bandaranayake.

**Methodology:** Bhagya M. Hathurusinghe.

**Resources:** Pradeepa C. G. Bandaranayake.

**Software:** Bhagya M. Hathurusinghe.

**Supervision:** D. K. N. G. Pushpakumara, Pradeepa C. G. Bandaranayake.

**Validation:** Bhagya M. Hathurusinghe, D. K. N. G. Pushpakumara.

**Visualization:** Bhagya M. Hathurusinghe, D. K. N. G. Pushpakumara, Pradeepa C. G. Bandaranayake.

**Writing – original draft:** Bhagya M. Hathurusinghe, Pradeepa C. G. Bandaranayake.

**Writing – review & editing:** Bhagya M. Hathurusinghe, D. K. N. G. Pushpakumara, Pradeepa C. G. Bandaranayake.

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
