## [Decision Letter · Decision Letter 0]

17 Aug 2022

PONE-D-22-19414Macroscopic and microscopic study on floral biology and pollinationof Cinnamomum verum Blume (Sri Lankan)PLOS ONE

Dear Dr. Bandaranayake,

Thank you for submitting your manuscript to PLOS ONE. After careful consideration, we feel that it has merit but does not fully meet PLOS ONE’s publication criteria as it currently stands. Therefore, we invite you to submit a revised version of the manuscript that addresses the points raised during the review process.

We look forward to receiving your revised manuscript.

Kind regards,

Muhammad Imran

Academic Editor

PLOS ONE

Journal Requirements:

"The Ministry of Primary Industries and Social Empowerment through the National Science Foundation of Sri Lanka under the special Cinnamon project – Grant No: NSF SP/CIN/2016/01. "

"Funded by the Ministry of Primary Industries and Social Empowerment through the National Science Foundation of Sri Lanka under the special Cinnamon project – Grant No: NSF SP/CIN/2016/01."

"The Ministry of Primary Industries and Social Empowerment through the National Science Foundation of Sri Lanka under the special Cinnamon project – Grant No: NSF SP/CIN/2016/01. "

Additional Editor Comments:

As the manuscript has been precisely reviewed by the experts in the disciplines and after thoroughly going through their reports it is cleared that there are some shorts comes which need to be address before publishing it so the decision is recorded as a Major Revision. Enclosed please find the reviewers’ report on your paper. I suggest to the authors a complete typographical and grammatical revision. Most of the references used to seem a little old to me, I suggest that the authors update and insert new references, to corroborate the assertions presented by the authors. Statistical section is very weak and need major attention in this section. So, according to the recommendations made by the valuable reviewers, the manuscript needs comprehensive major revisions.

Reviewers' comments:

Reviewer's Responses to Questions

**Comments to the Author**

1. Is the manuscript technically sound, and do the data support the conclusions?

Reviewer #1: Yes

Reviewer #2: No

2. Has the statistical analysis been performed appropriately and rigorously? 

Reviewer #1: No

Reviewer #2: No

3. Have the authors made all data underlying the findings in their manuscript fully available?

Reviewer #1: Yes

Reviewer #2: No

4. Is the manuscript presented in an intelligible fashion and written in standard English?

Reviewer #1: No

Reviewer #2: No

5. Review Comments to the Author

Reviewer #1: The manuscript presented is of good importance and will be benefit for the scientific community.

Here are my comments and suggestions:

The abstract should be rewrite and authors should present clearly this section to capture their result and how this will be useful.

No proper flow in the introduction and too difficult for the readers to understand the authors.

Material and method need adjustment: I will invite authors to summarize in a table format all the data capture, code use, period of collection and nature of the data (qualitative/quantitative). The analysis section very weak and authors need to address this points (which software was used for the analysis, which model was used, how the factors were considered/radom/fixed) in addition to this authors should add the regression analysis to display how the weather parameters impact on the flower both males and females.

Did authors test the pollen viability before conduction hand pollination. if not how sure authors relies on the current results.

In the results section, some sections look more like discussion. please move those sentences into the discussion section.

Please use correlation analysis considering only the quantitative variables to see the relationship among the variables (you can use as well the the variety type as factor see this link https://r-graph-gallery.com/199-correlation-matrix-with-ggally.html

In table 1, add the result of the LSD for mean separation.

The discussion section needs to be adjusted, only discuss the major discovery and how this will be used to improve the production. Also check for english typo across the document

Reviewer #2: The submitted manuscript does not have good theoretical bases on plant reproductive biology as it should have. The presented objectives are very unclear, the methods are not adequate, and the results does not support the conclusions.

6. PLOS authors have the option to publish the peer review history of their article (what does this mean?). If published, this will include your full peer review and any attached files.

Reviewer #1: No

Reviewer #2: No

---

## [Author Response · Author response to Decision Letter 0]

10 Oct 2022

Reviewers' comments:

Reviewer's Responses to Questions

Comments to the Author

Reviewer #1: The manuscript presented is of good importance and will be benefit for the scientific community.

Here are my comments and suggestions:

1. The abstract should be rewrite and authors should present clearly this section to capture their result and how this will be useful.

The abstract was revised following the instructions of the reviewer. 

2. No proper flow in the introduction and too difficult for the readers to understand the authors.

The introduction was revised in order to make it easy for the reader to understand

3. Material and method need adjustment: I will invite authors to summarize in a table format all the data capture, code use, period of collection and nature of the data (qualitative/quantitative). 

The data was included in the table (after the figure legends) as suggested by the reviewer. 

4. The analysis section very weak and authors need to address this points (which software was used for the analysis, which model was used, how the factors were considered/radom/fixed) in addition to this authors should add the regression analysis to display how the weather parameters impact on the flower both males and females.

The suggested analysis methodology was included in the methodology section line number 215-230.

The regression analysis suggested was included in table 3 and as well as Fig 8. 

5. Did authors test the pollen viability before conduction hand pollination. if not how sure authors relies on the current results.

Thank you for the suggestion. However, since we have sure results from scanning electron microscopy and also from the fruiting 

6. In the results section, some sections look more like discussion. please move those sentences into the discussion section.

Results section was adjusted. Sections similar to discussion was moved to discussion section.

7. Please use correlation analysis considering only the quantitative variables to see the relationship among the variables (you can use as well the the variety type as factor see this link https://r-graph-gallery.com/199-correlation-matrix-with-ggally.html

Thank you for your suggestion. The correlation analysis was carried out for quantitative variables using Minitab vr: 7 software. The above mentioned software was not used for the analysis. Included as Figure 11. 

8. In table 1, add the result of the LSD for mean separation.

The Values are added with means ± SD of all the female and male stage flowers in Sri Gemunu and Sri Wijaya (n) =10 replicates, Means denoted by the different letters within the same column are significantly different at P<0.05 (Table 1)

9. The discussion section needs to be adjusted, only discuss the major discovery and how this will be used to improve the production. Also check for english typo across the document

Discussion was adjusted as instructed by the reviewer. English typos were also corrected. 

Reviewer #2: The submitted manuscript does not have good theoretical bases on plant reproductive biology as it should have. The presented objectives are very unclear, the methods are not adequate, and the results does not support the conclusions.

Thank you for the review. The objectives, methods, data analysis and results were revised as suggested. 

1. The objetive presented here is very vague.

The objective in the introduction was rewritten

2. The atributes analysed do not characterize the resource accessibility. Besides, why do the authors need to characterize such resource? Are they flowers resources for pollinators? it was not explained.

The explanation for the analysis was explained in line 123-124 as reviewer suggested. 

3. Which data was tested and analysed?

The data analysis section was rewritten with broad explanation number 215-230.

---

## [Decision Letter · Decision Letter 1]

31 Oct 2022

Macroscopic and microscopic study on floral biology and pollinationof Cinnamomum verum Blume (Sri Lankan)

PONE-D-22-19414R1

We’re pleased to inform you that your manuscript has been judged scientifically suitable for publication and will be formally accepted for publication once it meets all outstanding technical requirements.

Kind regards,

Muhammad Imran

Academic Editor

PLOS ONE

Additional Editor Comments (optional):

I have thoroughly review the manuscript, all the comments from previous review process have been incorporated as mention our valuable reviewers. Now the manuscript is okay for publication and the decision is Accepted.

Reviewers' comments:

Reviewer's Responses to Questions

**Comments to the Author**

1. If the authors have adequately addressed your comments raised in a previous round of review and you feel that this manuscript is now acceptable for publication, you may indicate that here to bypass the “Comments to the Author” section, enter your conflict of interest statement in the “Confidential to Editor” section, and submit your "Accept" recommendation.

Reviewer #1: All comments have been addressed

2. Is the manuscript technically sound, and do the data support the conclusions?

Reviewer #1: Yes

3. Has the statistical analysis been performed appropriately and rigorously? 

Reviewer #1: Yes

4. Have the authors made all data underlying the findings in their manuscript fully available?

Reviewer #1: Yes

5. Is the manuscript presented in an intelligible fashion and written in standard English?

Reviewer #1: Yes

6. Review Comments to the Author

Reviewer #1: authors have. addressed all my comments and the manuscript can be accepted in the current form.

English spelling

7. PLOS authors have the option to publish the peer review history of their article (what does this mean?). If published, this will include your full peer review and any attached files.

Reviewer #1: **Yes: **Paterne AGRE

---

## [Editor Report · Acceptance letter]

15 Nov 2022

PONE-D-22-19414R1 

Macroscopic and microscopic study on floral biology and pollination of *Cinnamomum verum* Blume (Sri Lankan) 

Dear Dr. Bandaranayake:

I'm pleased to inform you that your manuscript has been deemed suitable for publication in PLOS ONE. Congratulations! Your manuscript is now with our production department. 

Kind regards, 

on behalf of

Dr. Muhammad Imran 

Academic Editor

PLOS ONE